# Insights into Bioactive Peptides in Cosmetics

**Le Thi Nhu Ngoc** [1] , **Ju-Young Moon** [2,*] **and Young-Chul Lee** [3,*]

1    Department of Nano Science and Technology Convergence, Gachon University, 1342 Seongnam-Daero, Sujeong-gu, Seongnam-si 13120, Republic of Korea; nhungocle92@gmail.com
2    Department of Beauty Design Management, Han-sung University, 116 Samseongyoro-16gil, Seoul 02876, Republic of Korea
3    Department of BioNano Technology, Gachon University, 1342 Seongnam-daero, Sujeong-gu, Seongnam-si 13120, Republic of Korea
*    Correspondence: bora7033@naver.com (J.-Y.M.); dreamdbs@gachon.ac.kr (Y.-C.L.); Tel.: +82-31-776-2863 (J.-Y.M.); +82-31-750-8751 (Y.-C.L.); Fax: +82-31-776-2864 (J.-Y.M. & Y.-C.L.)

**Abstract:** Bioactive peptides have gained significant attention in the cosmetic industry due to their potential in enhancing skin health and beauty. These small protein fragments exhibit various biological activities, such as antioxidant, anti-aging, anti-inflammatory, and antimicrobial activities, making them ideal ingredients for cosmetic formulations. These bioactive peptides are classified into four categories: signal, carrier, neurotransmitter-inhibitory, and enzyme-inhibitory peptides. This review provides insight into applying bioactive peptides in cosmetics and their mechanisms of action (e.g., downregulating pro-inflammatory cytokines, radical scavenging, inhibiting collagen, tyrosinase, and elastase synthesis). The abundant natural origins (e.g., animals, plants, and marine sources) have been identified as primary sources for extractions of cosmetic peptides through various techniques (e.g., enzymatic hydrolysis, ultrafiltration, fermentation, and high-performance liquid chromatography). Furthermore, the safety and regulatory aspects of using peptides in cosmetics are examined, including potential allergic reactions and regulatory guidelines. Finally, the challenges of peptides in cosmetics are discussed, emphasizing the need for further research to fully harness their potential in enhancing skin health. Overall, this review provides a comprehensive understanding of the application of peptides in cosmetics, shedding light on their transformative role in developing innovative and effective skincare products.

**Keywords:** classification of peptides; mechanisms of action; natural sources; safety assessments and challenges





## 1. Introduction

There is no denying that cosmetics have become an essential part of our daily routine, especially among women, and the cosmetic market, especially for natural ingredient-based products, is growing rapidly with a great demand for appearance improvement from consumers. Particularly, the development of novel cosmetic formulations based on bioactive compounds (e.g., antioxidants, proteins, peptides, and growth factors) with therapeutic and protective functions that can provide outstanding effects on human skin such as skin whitening, skin moisturizing, and skin rejuvenation, has quickly expanded [1].

Peptides are short chains of two to fifty amino acids linked together by peptide bonds [1–3]. Amino acids are the building blocks of proteins, and when they are joined in a chain, they form a peptide [1,2]. Peptides are smaller than proteins and typically composed of up to 50 amino acids, although specific thresholds can vary [1–3]. Among the prominent bioactive compounds, peptides have attracted the attention of scientists due to their ideal functionalities, especially as regulating/signaling molecules of homeostasis, stress, immunity, defense, growth, and reproduction [1–3], and their other strengths such as high safety, hypo-allergenicity, and their cost-effective production [1]. Derived from

various natural sources such as plants, animals, and microorganisms, these peptides have demonstrated diverse physiological effects, including antioxidant, anti-aging, moisturizing, collagen-stimulating, and wound-healing properties, which have been confirmed through several in vitro/in vivo pieces of evidence as well as clinical trial outcomes [1]. For instance, palmitoyl pentapeptide-3 was one of the first synthetic bioactive peptides used to stimulate collagen synthesis for anti-aging and wound healing treatments [4]. The copper Gly–His–Lys (Cu-GHK) was developed and incorporated into cosmetic products to heal skin, promote collagen synthesis, and repair DNA damage [5,6]. Acetyl hexapeptide-3 (Argireline®) is another popular commercialized peptide with potential anti-wrinkle and moisturizing properties [7].

Despite the growing scientific evidence supporting their potential application in cosmetics, there are only a handful of literature reports describing findings regarding promising applications. For instance, Ferreira et al. (2020) described the utilization of peptides for anti-aging treatment, but did not include treatment of all skin conditions [8]. Mazurkiewicz-Pisarek et al. (2023) reported applying antimicrobial peptides only for pharmaceutical, biomedical, and cosmeceutical applications [9]. Therefore, documenting the latest research and advancements in a review is necessary to provide comprehensive knowledge that can inspire further innovation in the formulation of peptide-based cosmetic products. In addition, there is a growing consumer demand for safe and effective cosmetic products with ingredients of natural or biological origin. Educating consumers about the potential of bioactive peptides through a well-written paper can help them make informed choices when selecting cosmetic products. It can be seen that understanding the potential of bioactive peptides in cosmetics can have broader implications beyond the cosmetic industry. Therefore, this review aims to explore the vast potential of bioactive peptides for cosmetic applications, shedding light on their intracellular mechanisms of action, their classifications, and their natural origins. It will delve into their multifaceted roles in skin health and beauty, addressing the underlying scientific principles and presenting relevant empirical evidence from both in vitro and in vivo studies. Moreover, it will discuss the safety assessment and challenges associated with using bioactive peptides in cosmetic formulations.

## 2. Mechanisms and Classification of Bioactive Peptides

### 2.1. Classification of Cosmetic Peptides

It has been demonstrated that bioactive peptides can exert their biological as well as cosmetic functions in different ways; thus, they are commonly classified into four categories according to their most outstanding features which includes signal peptides, carrier peptides, neurotransmitter-inhibitory peptides, and enzyme-inhibitory peptides (Figure 1 and Table S1) [4,10].

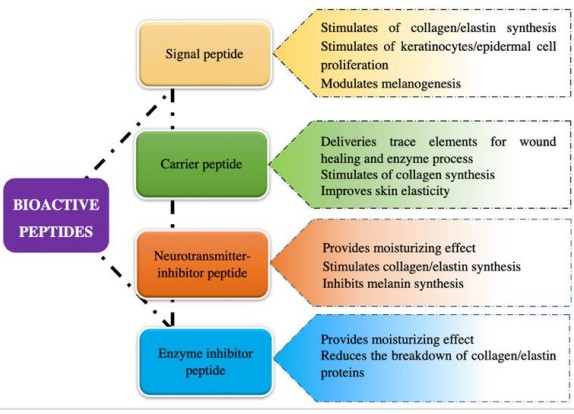

**Figure 1.** Classification of bioactive peptides based on cosmetic properties.

### 2.1.1. Signal Peptides

Signal peptides are active compounds that can prevent aging by stimulating skin fibroblasts, resulting in increased biological responses such as collagen, elastin, fibronectin, glycosaminoglycan, and proteoglycan production (Table S1) [10]. They may act as growth factors to activate protein kinase C which is mainly responsible for cell growth and migration [10].

One of the first cosmetic signal peptides is the palmitoyl peptide (Pal–Lys–Thr–Thr–Lys–Ser), which shows collagen modulating capabilities for anti-wrinkle and wound healing [4,10]. It is a sub-fragment of the carboxyl terminal pro-peptide of type I collagen that can dramatically enhance extracellular matrix production in fibroblasts [11,12], thereby effectively stimulating collagen (I, II, and III) and fibronectin production. Aruan et al. (2023) conducted a double-blind, split-face, placebo-controlled, and left-right randomized trial with 21 female subjects to assess the clinical efficacy of anti-wrinkle cream containing the peptide [12]. The result showed that in the course of a 12-week clinical trial, topical application of 3 ppm palmitoyl peptide reduced facial wrinkles/fine lines [12]. Another signal peptide that stimulates collagen synthesis is palmitoyl tripeptide-5 (palmitoyl–Lys–Val–Lys), which can mimic the effect of thrombospondin-1, a naturally occurring molecule that causes the sequence Lys–Arg–Phe–Lys to bind to the inactive transforming growth factor-$\beta$ (TGF-$\beta$), consequently, promoting the release of the active form of TGF-$\beta$ [13]. Thereafter, activated TGF-$\beta$ causes a constant increase in the amount of type I and III collagen produced in dermal fibroblasts [13]. A number of studies have demonstrated that palmitoyl tripeptide-3/5 enhances collagen synthesis and reduces collagen breakdown by interfering with MMP-1 and MMP-3 collagen degradation, leading thereby to improvements in aging signals [10,13]. For example, a controlled trial was conducted on 60 Chinese volunteers treated with palmitoyl pentapeptide-5 (2.5%) cream compared with a placebo cream (for 84 days and applied twice daily) [13]. It was confirmed that palmitoyl tripeptide-5 significantly reduced skin roughness, exhibiting a greater anti-wrinkle efficacy than the placebo or pal-KTTKS-containing creams [13]. Other commercialized signal peptides modulating collagen synthesis are described in terms of their anti-aging properties and mechanisms in Table S1.

Regarding the enhancement of elastin contents in the skin, several signal peptides (e.g., dipeptide-2/valy tryptophan, Val–Gly–Val–Ala–Pro–Gly, and palmitoyl oligopeptide) have been developed to stimulate elastin synthesis, leading to improved skin aging signals. For instance, hexapeptide Val–Gly–Val–Ala–Pro–Gly and its modified sequence palmitoyl hexapeptide-12 are highly specific to elastin molecules that stimulate collagen and elastin fibroblasts, as well as develop glycosaminoglycans and fibronectin [10]. The intracellular mechanism refers to the way in which they can reduce the production of proinflammatory mediators (e.g., IL-1, IL-6, and IL-8) and ultimately slow down the skin matrix's degradation [13]. Another signal peptide, palmitoyl oligopeptide, which contains such an elastin fragment, has been incorporated into cosmetic products to promote the proliferation of collagen, elastin, and hyaluronic acid, a role which suggests "reconstruction of the dermis" and "chemotaxis for restructuring and repair" properties [10]. Hahn et al. (2016) produced an anti-aging facial cream containing 1% palmitoyl oligopeptide, *Silybum marianum* seed oil, vitamin E, and other functional ingredients to combat facial wrinkles [14]. After 4 weeks of application, the volunteers' crow's feet wrinkles were reduced by 14.07% compared to pre-application, and the skin elasticity was observed to have increased by 8.79%. They were able to confirm, therefore, that a blend of palmitoyl oligopeptide and other cosmetics has a beneficial effect on facial wrinkles, elasticity, and skin tone.

### 2.1.2. Carrier Peptides

Carrier peptides have been designed to deliver essential wound healing cofactors for enzymatic processing and wound repair (Table S1) [10,15]. The first commercialized carrier

peptide was designed to deliver copper, a trace element necessary for wound healing, into the wounded tissue.

Copper is not only an essential trace element for wound healing but also a cofactor for enzymes lysyl oxidase, superoxide dismutase, and tyrosinase, which are important for collagen synthesis, superoxide dismutation (antioxidant action), and melanogenesis, respectively [5,10]. The first copper tripeptide Cu–GHK, potentially performs a role in the extracellular matrix by promoting regular collagen, elastin, glycosaminoglycan, and proteoglycan synthesis. This leads to the stimulation of cellular regulation molecules, and the regeneration and healing of skin and other tissues [5]. In particular, Siméon et al. (2000) demonstrated that Cu–GHK effectively promotes MMP-2 synthesis in skin fibroblasts, as represented by an increased expression of MMP-2 mRNA and the secretion of TIMP-1 and TIMP-2, resulting in fibroblast wound healing [5]. Pickart et al. (2015) demonstrated that Cu–GHK treatment remarkably reduced TNF-$\alpha$ levels induced by cytokines IL-6 and increased expression of various DNA repair genes [6]. It can be concluded that Cu–GHK contributes to a beneficial anti-aging effect through a number of mechanisms, especially by promoting regeneration, healing, and repair of damage. In fact, many studies have confirmed the clinical effects of Cu–GHK as a functional ingredient. Liu et al. (2023) tested the anti-wrinkle activity of a Cu–GHK formulation to improve skin elasticity, skin moisture, and skin-smoothing by enhancing collagen synthesis, thereby diminishing facial wrinkles and fine lines [16].

Another transition metal, manganese, is an essential nutrient involved in amino acid, cholesterol, antioxidant protection, and carbohydrate metabolism [17]. In addition, manganese-superoxide dismutase is considered to be very important in the defense against UV-induced photoaging [17,18]. It has been found that the level of manganese-superoxide dismutase is increased through the action of inflammatory mediators (e.g., IL-1 and TNF-$\alpha$) during UV irradiation [18]. Therefore, manganese tripeptide-1 (Mn–GHK) was formulated to provide a similar functionality to that of Cu–GHK in the photoaging treatment. Hussain and Goldberg (2007) evaluated the effects of a Mn–GHK-containing facial serum formulation in the treatment of various signs of photodamage for 12 weeks [17]. At the end of the treatment, volunteers noted an improvement in the appearance of cutaneous photodamage signs, the photodamage ranking having changed from moderate to mild, as well as the lack of any side effects such as inflammation [17].

### 2.1.3. Neurotransmitter-Inhibitor Peptides

Some of the most common signs of aging (e.g., wrinkles and fine lines) have also been controlled through strategies of muscle contraction regulated by neurotransmitters released from neurons through the use of neurotransmitter-inhibitor peptides [13,15]. In particular, the muscle contraction process occurs when vesicles containing the neurotransmitter acetylcholine join the neuron in order to break into two separate fragments including the vesicle and acetylcholine (ACh)-a neurotransmitter of the parasympathetic nervous system [13,15]. The vesicle is captured with SNARE complexes (soluble N-ethylmaleimide-sensitive factor activating protein receptor) and then fused with the neuron membrane, while the ACh is released in neuromuscular junctions between the muscle and the nerve [19]. The released ACh binds to the acetylcholine receptors that are present on the muscle cell's surface, leading to muscle contraction. It is reported that the entire process is regulated by SNAP-25, a receptor protein present in the neuronal membrane, which is associated with the vesicle and directly regulates binding with the SNARE complex as well as the fusion with the membrane of the vesicle [19]. According to the contraction mechanism, a number of peptides have been developed with sequences similar to SNAP-25 proteins, which can compete for the binding sites of SNARE complexes, leading to their structural instability and inhibition of the release of ACh at the neuromuscular junction, inducing muscle relaxation. These synthetic peptides (e.g., acetyl hexapeptide-3, pentapeptide-3, pentapeptide-18, and tripeptide-3) exhibit specific neuro-suppressive abilities; accordingly, they are called neurotransmitter-inhibitor peptides [19] (Table S1).

One of the most popular commercialized neurotransmitter-inhibitor peptides is acetyl hexapeptide-3 (Argireline®), offering advanced properties in anti-wrinkle and moisturizing effects. This peptide, which is similar to botulinum toxin, is able to mimic the N-terminal end of the SNAP-25 protein and compete for a site in the SNARE complex, resulting in destabilization of its formation as well as inhibition of ACh release and, eventually, decreased muscle contraction [7,15,20]. A clinical study by Ruiz et al. (2010) investigated the anti-wrinkle benefits of an oil in water (O/W) emulsion containing acetyl hexapeptide-3 in 20 human subjects for 30 days of topical application. It showed positive signs with a reduction in wrinkle depth and size by 59% and 41%, respectively, compared with the placebo control [21].

### 2.1.4. Enzyme-Inhibitory Peptides

Enzyme-inhibitory peptides can directly or indirectly inhibit enzymes that break down collagen and other proteins and interfere with that process. A number of enzyme-inhibitory peptides such as soy oligopeptides, rice-derived peptides and silk fibroin peptides have been used as inactive ingredients for skincare products (Table S1) [15,22].

Soybean-derived peptides, comprised of 3–6 amino acids, possess various biological activities including antioxidative, blood-lipid-lowering, and blood-pressure-lowering effects. These biological properties show an overwhelming effect in increasing levels of proapoptotic Bcl-2 protein and decreasing cyclobutene pyrimidine dimer-positive cells, apoptotic cells, expression of Bax and p53 proteins in the epidermis due to UVB irradiation [15,22]. As a result, they are frequently used as anti-aging agents, skin moisturizers, hair growth promoters, and cleaning detergents. A pseudo-randomized clinical study of ten Caucasian women, confirmed the superiority of soybean peptide (2%) emulsion in increasing the amount of extracellular matrix components such as collagen and glycosaminoglycan contents [23].

Rice-derived peptides (molecular weight < 300 Da) obtained after special processing of rice-bran protein, greatly inhibit activity of MMPs and stimulate the expression of hyaluronan synthase 2 genes in human keratinocytes cells [24]. Manosroi et al. (2012) successfully produced formulas containing niosomes encapsulated in rice-bran peptides, and demonstrated that they have ideal clinical anti-aging properties [25].

Silk fibroin peptide is obtained from the silkworm *Bombyx mori*. This peptide is able to inhibit inflammation, specifically by increasing the anti-inflammatory activity of tTAT-superoxide dismutase, which has been reported to effectively penetrate into skin cells and tissues and to exert anti-oxidant effects in an inflamed-mouse model [26].

### 2.2. Mechanisms of Action

Bioactive peptides, with their powerful single/multifunctional biological properties (e.g., antimicrobial, antioxidant, anti-aging, and anti-inflammatory activities) have been widely applied as functional ingredients in the dermatology and cosmetology fields. The peptides can improve skin health in a number of aspects, including extracellular matrix synthesis, innate immunity, inflammation, and pigmentation [1,27–30]. The detailed mechanisms of each cosmetic property are fully described in Table 1 and Figure 2.

**Table 1.** Intracellular mechanisms of cosmetic properties.

| Cosmetic Properties | Mechanism | Effective Factors |
|---|---|---|
| Antioxidant activity | Prevents the deleterious effects of oxidative stress caused by overproduction of ROS in the skin [31]<br>Act as antioxidants through hydrogen atom transfer, single electron transfer, and chelating transition pro-oxidant metals [31] | Antioxidant properties depend on their structural properties: molecular weight, hydrophobicity, and amino acid sequence (Pro, His, Cys, Phe, Try, and Tyr) [28]<br>Peptides with lower molecular weight show effective antioxidant properties [32] |
| Anti-inflammatory activity | Possesses anti-inflammatory capacity mediated by the inhibition and induction of the immune systems in cell lines [33]<br>Downregulates pro-inflammatory mediators (e.g., TNF-$\alpha$, IL-1$\alpha$, IL-1$\beta$, IL-2, IL-6, IL-8, IL-12, and IFN-$\gamma$ receptors) and regulates immune system [33] | Anti-inflammatory activity is related to their ability to bind to the lipid A moiety of lipopolysaccharides (LPS) and interference with LPS-CD14 interactions by competing with the LPS-binding peptide [34] |
| Antimicrobial activity | Provides antimicrobial activity based on membrane lytic mechanisms whereby peptides can directly affect cell membrane integrity through the formation of transmembrane channels, resulting in cytoplasm leakage and cell death [35]<br>Involved with the inhibition of intracellular activities in nucleic acid, protein and cell-wall synthesis, protein folding, lipopolysaccharide formation, and cell-division progress [35]<br>Induces a loss of regulated iron transport, leading to membrane permeation and DNA damage, and subsequently to bacterial destruction [1] | Peptides with cationic charge (from +2 to +9) show a strong ability to interact with the negatively charged membranes of microorganisms [36] |
| Anti-aging properties | Collagenase inhibition<br>Inhibits mitogen-activated protein kinase (MAPK) and nuclear factor $\kappa$B (NF-$\kappa$B) signaling pathways, and histone modification [37]<br>Suppresses the activities and expressions of MMP by elevating tissue inhibitors of matrix metalloproteinases (TIMP) levels and blocking activation of MAPK signaling pathway [38] | Low molecular weight peptides (<1 kDa) possess higher inhibitory activity against MMP, p-JNK, p-p38, and p-ERK in MAPK signaling pathways than that of larger molecular weight peptides [39] |
| | Hyaluronidase inhibition<br>Inhibits the degradation of hyaluronic acid for protecting skin [40] | Hyaluronidase inhibitor capacity depends on molecular weights as follows: large molecular peptides (3–10 kDa) > medium molecular peptides (1–3 kDa) > small molecular peptides (<1 kDa) [41] |
| | Tyrosinase inhibition<br>Blocks the active site or chelates copper ions of tyrosinase to inhibit tyrosinase activity [42]<br>Downregulates the activation of microphthalmia-associated transcription factor (MITF), an important event during melanogenesis, to suppress melanin synthesis [43]<br>Downregulates cAMP signaling pathway as an anti-melanogenic activity to inhibit melanin synthesis [1,42,43] | Peptides consisting of amino acids with hydroxyl groups (Ser and Thr), aliphatic amino acid residues (Val, Ala, and Leu), and hydrophobic compounds exhibit great tyrosinase inhibitory activities [44]<br>Peptides with molecular weight < 3 kDa show higher tyrosinase activity that that of the whole collagen hydrolysate [1] |
| | Elastase inhibition<br>Downregulates the activation of elastase enzyme to protect mechanical properties of skin tissues that are impaired by overproduction of the enzyme elastase [45] | --- |

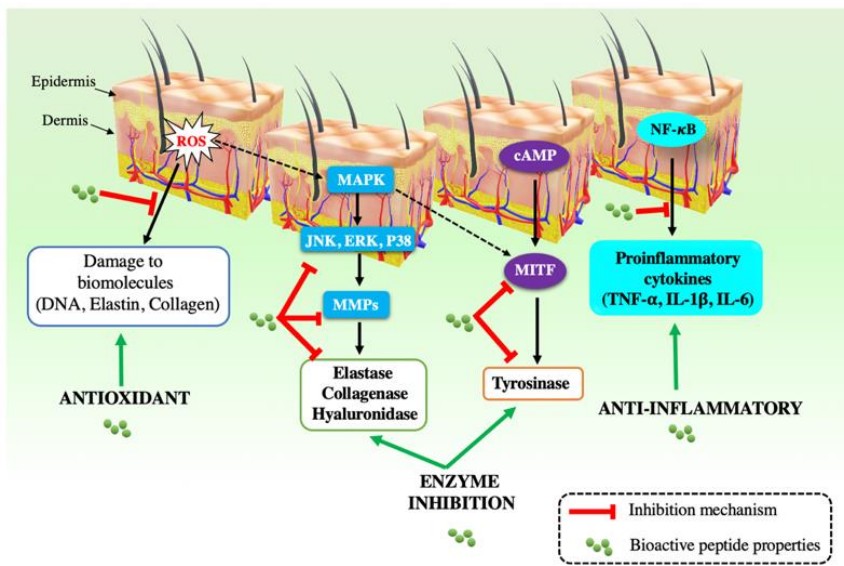

**Figure 2.** Schematic summary of the mechanism of bioactive peptides related to their potential cosmetic application. ROS: Reactive oxygen species; MAPK: mitogen-activated protein kinases; ERK: Extracellular-signal-regulated kinases; JNK: c-jun N-terminal kinases; MMPs: matrix metalloproteinases; cAMP: cyclic adenosine monophosphate; MITF: microphthalmia-associated transcription factor; TNF-α: tumor necrosis factor-α; IL-1β: interleukin-1β; IL-6: interleukin-6 [1].

## 3. Natural Sources of Bioactive Peptides

Bioactive peptides with promising cosmetic properties come from different sources, including chemical synthesis and natural sources. Chemical synthesis involves using a mixture of amino acids as a starting material, allowing peptides of different amino acid sequences and combinations to be obtained. Natural sources, such as plants, animals, and marine sources, can be used to extract bioactive peptides using various approaches (e.g., enzymatic hydrolysis, microbial fermentation, chemical digestion, high-performance liquid chromatography (HPLC), electro-membrane fractionation by electrodialysis with ultrafiltration, cell filtration chromatography, and recombinant production). The diversity of natural sources is expected to produce a wide range of desirable functional peptide structures. Therefore, this study focuses on the natural sources for deriving bioactive peptides for cosmetic applications.

### 3.1. Plant Sources

Plants are well known as a rich source of proteins without saturated fatty acids, that can carry useful ingredients and can perform several functions in humans (e.g., antidiabetic, immunomodulatory, antihypertensive, antimicrobial, and antioxidant activities) [46]. Especially bioactive peptides with antioxidant and antimicrobial activities are expected to provide great cosmetic benefits (Table 2). Common antimicrobial peptides comprise thionins, defensins, 2S albumin-like proteins, cyclotides, and lipid transfer proteins, of which thionins are the first identified to play a significant role in protecting plants against invading bacteria [47]. The antimicrobial properties help peptides penetrate the bacterial membrane, then create pores and eliminate the bacteria by altering their homeostasis. Additionally, plant-derived peptides are also toxic to a variety of Gram-negative and Gram-positive bacteria, and even mammalian bacteria [47]. For example, Zhang et al. (2019) purified four antioxidant peptides from alcalase hydrolyzed soybean (*Glycinemax* L.) using gel filtration chromatography and reversed-phase HPLC [48]. Four sequences (Val –Val–Phe–Val–Asp–Arg–Leu, Val–Ile–Tyr–Val–Val–Asp–Leu–Arg, Ile–Tyr–Val–Val–Asp–Leu–Arg, and Ile–Tyr–Val–Phe–Val–Arg) were obtained and displayed the desired DPPH and ABTS radical-scavenging activities, oxygen radical absorbance capacity (ORAC), and ferric reducing antioxidant power (FRAP). Moreover, Ile–Tyr–Val–Val–Asp–Leu–Arg

and Ile–Tyr–Val–Phe–Val–Arg significantly upregulated total reduced glutathione synthesis, inhibited ROS-mediated inflammatory responses, and also enhanced catalase and glutathione reductase activities, therefore, these soybean-derived peptides are expected to be great cosmetic candidates [48].

### 3.2. Animal Sources

Animal-derived peptides are also increasingly attracting interest as prominent candidates in the field of cosmetology. Milk, meat, fish, and egg are usually the origins of peptide derivatives. It has been reported that peptides of animal origin can lower blood pressure, stimulate the immune system, inhibit proline-specific endopeptidase activity, induce smooth muscle contraction, have antibacterial and antimicrobial activities, and improve the nutritional value of foods. Firstly, milk proteins (e.g., donkey milk, buffalo milk, and goat milk) are considered the most important source of animal-derived peptides. For instance, Yang et al. (2020) successfully derived lactoferrin peptide from buffalo milk protein using purification and solid phase synthesis [49]. This peptide showed high antioxidant capacity through superoxide dismutase (SOD), malondialdehyde (MDA), and glutathione peroxidase (GSH-PX) activities [49]. Secondly, eggs, which account for about 11% of daily protein intake with high-quality proteins, are well recognized as promising sources of bioactive peptides. Peptides derived from egg proteins are mainly extracted using the enzymatic hydrolysis technique (e.g., trypsin, pepsin, chymotrypsin, thermolysin, and alcalase), and present several beneficial properties (e.g., antioxidant, antihypertensive, antimicrobial, anticancer, antidiabetic, and immunomodulatory activities) [46]. For example, the peptide $P_{21\text{-}3\text{-}75\text{-}B}$ purified from duck egg white protein via "SEEP-Alcalase" hydrolysis exhibited high antioxidant capacity and high nutritious value for daily intake [50]. Additionally, the potential of meat sources (e.g., pork, beef, chicken, mutton, and duck) that are high in nutrients (iron, vitamin B12, amino acids, and folic acid) should not be overlooked. The meat-based peptides possess several physiological activities such as antioxidative, antimicrobial, antithrombotic, and antihypertensive [51–53].

### 3.3. Marine Sources

Marine-derived peptides have also been confirmed to have high nutritional value and great cosmetic properties, which can provide both health and cosmetic benefits. Particularly, marine environments are more biologically diverse than terrestrial environments, and because of these organisms' unique adaptions to dark, cold, and high-pressure environments during their evolution, they are able to express different proteins to overcome these incompatible environments [46]. Marine-based peptides exhibit various bioactivities including antioxidant, neuroprotective, antidiabetic, immunomodulatory, antibacterial, antiproliferative, and antioxidant activities. For instance, the *Porphyra dioica* algae were used to extract eight peptide sequences through enzymatic hydrolysis and reverse-phase HPLC approaches [54]. Among the derived peptides, the Asp–Tyr–Tyr–Lys–Arg sequence exhibited the highest antioxidant activity and the Tyr–Leu–Val–Ala sequence showed dipeptidyl peptidase IV (DPP-IV) inhibitory activities, while the Thr–Tyr–Ile–Ala peptide had the highest angiotensin-converting enzyme (ACE) inhibitory activity [54]. It can be seen that these peptides have potential applications as ingredients to improve health and skin appearance [54]. Li et al. (2021) successfully purified an algicidal peptide—Malformin C—from the marine fungus *Aspergillus* species [55]. Malformin C showed dose-dependent antimicrobial activities against two strains of noxious red tide algae (*Chattonella marina* and *Akashiwo sanguinea*) [55]. In another study, a new cyclic lipopeptide aciculitin D was extracted from a *Poecillastra* species of marine sponge and showed cytotoxicity against HCT-116 and HeLa cancer cells with an $IC_{50}$ of 1.4 µM and 4.5 µM, respectively [56].

### 3.4. Edible Insect Sources

Edible insects have been considered as new sources of peptides based on their good source of protein. Insect-derived bioactive peptides are characterized by a variety of

properties such as antioxidant, anti-inflammatory, antidiabetic, antimicrobial, and ACE-inhibitory activities. The research on insect-derived peptides is relatively new. For instance, Zielinska et al. (2018) successfully identified antioxidant and anti-inflammatory peptides through stimulation of the gastrointestinal digestion of three edible insects (*Schistocerca gregaria*, *Tenebrio Molitor*, and *Gryllodes Sigillatus*) [57]. Souse et al. (2020) hydrolyzed edible insect *Alphitobius diaperinus* protein with two enzymes alcalase and corolase to extract bioactive peptides for food ingredients [58]. The derived peptide was observed to have high antioxidant and antihypertensive activities, but no antimicrobial or antidiabetic properties, suggesting that this peptide is a great supplement/ingredient in the food industry with promising health benefits [58].

**Table 2.** Bioactive peptides derived from natural sources for cosmetic applications.

| Classification of Peptides | Sources | Type of Bioactive Peptide Preparation | Peptide Sequences | Main Activities | References |
|---|---|---|---|---|---|
| Plants | Buckwheat (*Fagopyrum esculentum Moench.*) seed | Enzyme hydrolysis | Ala–Leu–Pro–Ile–Asp–Val–Ala–Asn–Ala–Tyr–Arg Thr–Asn–Pro–Asn–Ser–Met–Val–Ser–His–Ile–Ala–Gly Lys | Antimicrobial activity | [59] |
| | *Amaranthus hypochondiracus* seed | Reverse-phase high pressure liquid chromatography | Phe–Val–Pro–Asn–Gln–Asp–Glu–Val–Gln–Arg–Glu–Leu–Gln–Gln–Cys–Ile–Gln–Arg–Cys–Gln–Arg–Glu–Arg–Gly Gln–Met–Gly Gln–Met–Lys | Antimicrobial activity | [60] |
| | Mulberry (*Morus atropurpurea Roxb.*) Leaf | Neutrase-hydrolysate hydrolyzation using ion exchange chromatography, gel filtration chromatography, and reverse-phase HPLC | Ser–Val–Leu Glu–Ala–Val–Gln Arg–Asp–Tyr | Antioxidant activity | [61] |
| | *Jiupei* (fermented grains) | Fermentation, ultrafiltration, and reverse-phase HPLC | Val–Asn–Pro Tyr–Gly–Asp | Antioxidant activity | [62] |
| | Alcalase-hydrolyzed soybean (*Glycinemcax L.*) | Gel filtration chromatography and reverse-phase HPLC | Val–Val–Phe–Val–Asp–Arg–Leu Val–Ile–Tyr–Val–Val–Asp–Leu–Arg Ile–Tyr–Val–Val–Asp–Leu–Arg Ile–Tyr–Val–Phe–Val–Arg | Antioxidant, anti-inflammatory, and skin-whitening activities | [48] |
| | Chickpea (*Cicer arietinum L.*) | Ion-exchange chromatography, gel filtration chromatography, and reverse-phase HPLC | Leu–Thr–Glu–Ile–Pro | Antioxidant activity | [63] |
| | Defatted walnut (*Juglans regia L.*) | Enzymatic hydrolysis | Gln–Leu–Gln–Val–Leu–Arg–Pro–Arg Gln–Leu–Pro–Arg Val–Asn–Leu–Asn–Pro–His–Lys–Leu–Pro–Leu Leu–Gly Leu–Leu–Pro–Ser–Phe–Seu–Asn–Ala–Pro–Arg | Antioxidant activity | [64] |



**Table 2.** *Cont.*

| Classification of Peptides | Sources | Type of Bioactive Peptide Preparation | Peptide Sequences | Main Activities | References |
|---|---|---|---|---|---|
| Animals | *Arthrospira platensis* | Enzymatic hydrolysis | Gly–Met–Cys–Cys–Ser–Arg<br>Tyr–Gly–Phe–Val–Met–Pro–Arg–Ser–Gly<br>Trp–Phe–Arg | Antioxidant, hemolysis inhibition, and collagen-stimulating activities | [65] |
| | Lactoferrin or buffalo milk | Enzymatic hydrolysis and solid phase synthesis | Ser–Val–Asp–Gly–Lys–Glu–Asp–Leu–Ile–Trp | Antioxidant, superoxide dismutase (SOD), glutathione peroxidase (GSH-PX), and malondialdehyde (MDA) activities | [49] |
| | Hard cow milk cheese | Reverse-phase HPLC | Glu–Ile–Val–Pro–Asn<br>Asp–Lys–Ile–His–Pro–Phe<br>Lys–Ala–Val–Pro–Tyr–Pro–Gln<br>Val–Ala–Pro–Phe–Pro–Gln | Antioxidant and metal chelating activities | [66] |
| | Mastitic cow milk | Reverse-phase HPLC | Ile–Asp–Trp–Lys–Lys–Leu–Leu–Asp–Ala–Ala–Lys–Gln–Ile–Leu | Antimicrobial activity | [67,68] |
| | Goat milk | Fermentation | Ser–Ala–Glu–Glu–Gln–Leu–His–Ser–Met–Lys<br>Ile–Ala–Lys–Tyr–Ile–Pro–Ile–Gln–Tyr–Val–Leu–Ser–Arg<br>Glu–Ala–Leu–Glu–Lys–Phe–Asp–Lys | Antioxidant activity | [69] |
| | Bullfrog skin (*Rana catesbeiana* Shaw) | Enzymatic hydrolysis | Leu–Glu–Glu–Leu–Glu–Glu–Glu–Leu–Glu–Gly Cys–Glu | Antioxidant activity | [70] |
| | Chicken dark meat | Enzymatic hydrolysis | Tyr–Ala–Ser–Gly Arg | Antioxidant activity | [51] |
| | Sour pork meat | Fermentation | Glu–Ser–Thr–Val–Pro–Glu–Arg–Thr–His–Pro–Ala–Cys–Pro–Asp–Phe–Asn | Antioxidant capacity | [52] |

**Table 2.** *Cont.*

| Classification of Peptides | Sources | Type of Bioactive Peptide Preparation | Peptide Sequences | Main Activities | References |
|---|---|---|---|---|---|
| Marine | *Arthrospira platensis* (*Spirulina*) | Enzymatic hydrolysates | Ala–Asn–His–Gly Leu–Ser–Gly Asp–Ala–Ala–Val–Glu–Ala–Asn–Ser–Tyr–Leu– Asp–Tyr–Ala–Ile–Asn–Ala–Leu–Ser | Skin moisturizing activity | [71] |
| | Dunaliella salina | Ultrasound extraction and membrane ultrafiltration | Ile–Leu–Thr–Lys–Ala–Ala–Ile–Glu–Gly Lys Ile–Ile–Tyr–Phe–Gln–Gly Lys Asn–Asp–Pro–Ser–Thr–Val–Lys Thr–Val–Arg–Pro–Pro–Gln–Arg | Antioxidant activity | [72] |
| | Algae *Gracilariopsis lemaneiformis* | Enzymatic hydrolysis | Glu–Leu–Trp–Lys–Thr–Phe | Antioxidant activity | [73] |
| | Algae *Porphyra dioica* | Reverse-phase HPLC | Asp–Tyr–Tyr–Lys–Arg Thr–Tyr–Ile–Ala | Antioxidant and antimicrobial activities | [54] |
| | Algae *Porphyra yezpensis* | Ultrafiltration, molecular sieve chromatography, and ion exchange chromatography | Thr–Pro–Asp–Ser–Glu–Ala–Leu | Antimicrobial activity | [74] |
| | Fungi *Acremonium sp.* NTU492 | Enzyme hydrolysis | Gln–Ile–Ile–Ile–Val–Ile–Ile–Leu | Anti-inflammatory activity | [75] |
| | Fungi *Aspergillus allahabadii* and *A. ochraceopetaliformis* | Fermentation | Ala–Phe–Tyr–Pro–Leu–Val | Antimicrobial activity | [76] |
| | Fungi *Aspergillus* sp. | Ethanol extraction and HPLC | Cys–Cys–Val–Leu–Leu | Antimicrobial activity | [55] |
| | Sponge *Poecillastra* sp. | Ethanol extraction and HPLC | Abu–Thr–Tyr–Abu–Gly Thr–His | Antioxidant and high biological activities | [56] |
| Edible insects | *Schistocerca gregaria* | Gel filtration chromatography | Gly–Lys–Asp–Ala–Val–Ile–Val Ala–Ile–Gly Val–Ala–Ile–Glu–Arg Phe–Asp–Pro–Phe–Pro–Lys Tyr–Glu–Thr–Gly Asn–Gly–Ile–Lys | Antioxidant and anti-inflammatory activities | [57] |
| | *Alphitobius diaperinus* | Enzyme hydrolysis | Ala–Arg–Asn–Asp–Cys–Gln–Glu–His–Met–Phe– Thr–Trp–Val–Tyr | Antioxidant activity | [58] |
| | *Tenebrio molitor* | Gel filtration chromatography | Pro–Ala–Leu–Leu–Leu Ala–Ala–Gly–Ala–Pro–Pro Ser–Leu–Ala–Pro–Lys | Antioxidant activity | [77] |
| | *Bombyx mori* | Enzyme hydrolysis | Ser–Trp–Phe–Val–Thr–Pro–Phe Asn–Asp–Val–Leu–Phe–Phe | Antioxidant activity | [26] |

## 4. Safety Assessment of Peptides Used as Cosmetic Ingredients

The increased demand for cosmeceutical products has led to an interest in developing new-generation products based on bioactive peptides. Thus, safety assessment of cosmetic peptides, especially in terms of effectiveness, should be seriously considered. It is well known that a safe and effective amount is the amount of a composition or compound that is sufficient to produce significant positive skin benefits but small enough to avoid such undesired effects such as skin toxicity, sensitization, and irritation [78].

For instance, palmitoyl pentapeptide-4, the most commonly used peptide as an active ingredient in skincare, is rated for safety in a number of current skincare formulations [78]. Present skin care formulations can be manufactured with a combination of other optional ingredients known for safety assessment in cosmetic products, including emollients, vitamins, sugar amines, humectants, and sunscreen actives. The International Cosmetic Ingredient Dictionary and Handbook (INCI Dictionary, 2023) reported the safety of myristoyl pentapeptide-4 and palmitoyl pentapeptide-4 as used in cosmetic formulations [79]. It indicated that myristoyl pentapeptide-4 can be used at up to 0.05% in eye makeup preparations. Palmitoyl pentapeptide-4 is reported to be used at up to 0.0012% in eye lotions and face powder without irritation or sensitization.

The Personal Care Products Council (2018) survey of ingredient concentrations used in cosmetics reported that the safety assessments ranged from $1 \times 10^{-7}$% (palmitoyl tripeptide-1 and palmitoyl hexapeptide-12) to 0.002% (palmitoyl hexapeptide-12) [80]. Additionally, the data indicated that these peptides are being used in concentrations between 1 and 30 ppm, and their application at amounts lower than 10 ppm is customary [80]. In fact, a number of commercialized products have been reported to be safe when containing adequate amounts of bioactive peptides. For example, a trade-name formulation containing 100 ppm palmitoyl tripeptide-12 is non-irritating to the eyes of rabbits [80]. The result of an in vitro neutral red release assay indicates that a commercialized mixture containing 200 ppm palmitoyl hexapeptide-12 induces unimportant cytotoxicity [80]. On the other hand, excessive amounts of some peptides have been demonstrated to cause certain side effects. A trade name mixture containing 100 ppm palmitoyl tripeptide-1 is slightly irritating to the eyes of rabbits. Another trade-name formulation containing 500 ppm palmitoyl hexapeptide-7 is classified as a mild irritant [80].

The use of palmitoyl pentapeptide-3 and palmitoyl hexapeptide-14 for facial powders or spray cosmetics is also regulated at a maximum concentration of 0.06% [4]. Commercialized products named BIOPEPTIDE-CL and NANOFIBERGEL-CS containing 100 ppm palmitoyl oligopeptide and palmitoyl dipeptide-18, respectively, were nontoxic in acute oral toxicity studies in rats [4].

As reported by the FDA's Voluntary Cosmetic Registration Program (VCRP), acetyl hexapeptide-8 is reported to be used in 452 cosmetic products, including 244 leave-on and 30 rinse-off products [81]. Although it is used in concentrations up to 0.005% for eye lotions and face and neck goods, powder products can only contain up to 0.0001% of the total formulations, and a maximum amount of 0.000005% is used for rinse-off (skin cleansing) products [81]. In addition, the report reveals that acetyl hexapeptide-8 can be used as often as several times per day, and that daily or occasional use may extend over many years without serious side effects.

## 5. Challenges of Cosmetic Peptide Applications

The development of new generation peptide-based products not only brings several outstanding advantages but also faces various challenges in the cosmetics industry. Firstly, production conditions and techniques need to be carefully considered as well as improved to achieve high yield peptide contents with desirable bioactivity characteristics as well as multifunctional and potent activities. In addition, it is necessary to address the issue of maintaining the structural stability and bioactivity of bioactive peptides during the manufacture and storage of products. This issue is related to their properties being heavily influenced by various variables, including pH, temperature, interactions with other

active components, process formulation (e.g., concentration, packaging, encapsulation, and delivery of active ingredients), and particularly the distribution of active peptides [1]. In particular, the foremost challenge is delivery to the target, since most oligopeptides with high molecular weight and low lipophilicity are described as poor in osmosis, and so it is important to choose peptides with low molecular weight for enhanced efficiency of skin permeability [82,83]. Another limitation of peptides utilization is related to their hydrophobic properties as well as poor solubility at high concentrations; thus, a formulation containing a small amount of peptides may improve solubility while conferring the same level of bioactivity [1]. In addition, economic concerns about the mass production of peptides-based cosmetics have been raised due to the high cost of synthetically pure peptides. Although using low concentrations of expensive synthetic peptides can solve the economic and aqueous solubility problems, detection of ppm levels of peptides in a complex cream presents a tough challenge [82].

On the other hand, oral products should be carefully considered not only for their noted parameters but also for the bioavailability properties of their active compounds exhibiting both their physiological bioactivities and their structural integrity during digestion, transport, and intestinal absorption [1,84]. During oral administration, these peptides are digested in the gastrointestinal tract by exposure to the enzymes of the gastric and small intestinal brush border membrane and the acidic states of the stomach [84]. It is noted that the original peptides can be absorbed and passed into the systemic circulation only at nano-molar or pico-molar concentrations [84]. In addition, concerns about the bitter taste of some peptides intended for oral use necessitates their combination with other components into a suitable formulation to enhance the products' sensory properties [1].

## 6. Conclusions

In conclusion, exploring bioactive peptides for cosmetic applications has opened up new possibilities in skincare and beauty. This review has provided valuable insights into the potential of peptides as active ingredients in cosmetic formulations. Bioactive peptides extracted from various sources (plants, animals, marine sources, and edible insects) exhibit single or multifunctional cosmetic properties, including anti-aging, antioxidant, anti-inflammatory, and antimicrobial activities. Several in vitro/in vivo studies and clinical trials demonstrate that bioactive peptides are effective in enhancing skin appearance in both topical applications and oral administration, such as improving skin whitening and moisturizing and reducing skin aging based on various intracellular mechanisms. Safety and regulatory aspects have also been discussed, acknowledging the importance of evaluating potential allergic reactions and complying with regulatory guidelines. Consumer safety is paramount, and thorough testing and assessment must be conducted to ensure the suitability and tolerability of bioactive peptides in cosmetics. In addition, further studies should be conducted to unlock the full potential of bioactive peptides in cosmetics. This includes investigating novel sources of peptides, optimizing extraction methods, and exploring innovative delivery systems to enhance their penetration and activity within the skin. In conclusion, by harnessing their uniqueness to develop innovative and effective skincare products it is possible to cater to the diverse needs of consumers.

**Supplementary Materials:** The following supporting information can be downloaded at: https://www.mdpi.com/article/10.3390/cosmetics10040111/s1, Table S1: Classification of cosmetic peptides.

**Author Contributions:** L.T.N.N. planned the study and contributed the main ideas; L.T.N.N. was principally responsible for the writing of the manuscript; Y.-C.L. and J.-Y.M. commented on and revised the manuscript. All authors have read and agreed to the published version of the manuscript.

**Funding:** This work was supported by the Basic Science Research Program through the National Research Foundation of Korea funded by the Ministry of Education (2021R1F1A1047906) and by the Basic Science Research Capacity Enhancement Project through a Korea Basic Science Institute (National Research Facilities and Equipment Center) grant funded by the Ministry of Education (2019R1A6C1010016).

**Conflicts of Interest:** The authors declare no conflict of interest.

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
