# Peer review of "Insights into Bioactive Peptides in Cosmetics"

_cosmetics, doi:10.3390/cosmetics10040111_

Round 1

Reviewer 1 Report

This review is presented with all the informations.

latest references are missing. no work has been cited from 2023.

I suggest including classification of bioactive peptides first and then mechanism.

What about the safety and toxicity of these peptides? if possible, you can include.

Author Response

Response sheet

Manuscript ID: cosmetics-2516626
Type of manuscript: Review
Title: Insights into Bioactive Peptides in Cosmetics
Authors: Le Thi Nhu Ngoc, Ju-Young Moon*, Young-Chul Lee*

Reviewer 1

This review is presented with all the informations.

  1. The latest references are missing. No work has been cited from 2023.

Response: Thank you for your comment. The latest references have been cited in the upgraded manuscript.

  1. I suggest including the classification of bioactive peptides first and then the mechanism.

Response: Thank you for your suggestion. According to your comment, the structure of this manuscript has been rearranged. The section “Classification of bioactive peptides” has been arranged first and then the mechanism of actions.

  1. What about the safety and toxicity of these peptides? if possible, you can include.

Response: Thank you for your comment. The safety assessment of these peptides used as cosmetic ingredients has been inserted in the upgraded manuscript.

“4. Safety assessment of peptides used as cosmetic ingredients

The increased demand for cosmeceutical products has led to an interest in developing new-generation products based on bioactive peptides. Thus, safety assessment of cosmetic peptides, especially in terms of effectiveness, should be seriously considered. It is well known that a safe and effective amount is the amount of a composition or compound that is sufficient to produce significant positive skin benefits but small enough to avoid such undesired effects such as skin toxicity, sensitization, and irritation [78].

For instance, palmitoyl pentapeptide-4, the most commonly used peptide as an active ingredient in skincare, is rated for safety in a number of current skincare formulations [78]. Present skin care formulations can be manufactured with a combination of other optional ingredients known for safety assessment in cosmetic products, including emollients, vitamins, sugar amines, humectants, and sunscreen actives. The International Cosmetic Ingredient Dictionary and Handbook (INCI Dictionary, 2023) reported the safety of Myristoyl pentapeptide-4 and palmitoyl pentapeptide-4 as used in cosmetic formulations [79]. It indicated that Myristoyl pentapeptide-4 can be used at up to 0.05% in eye makeup preparations. Palmitoyl pentapeptide-4 is reported to be used at up to 0.0012% in eye lotions and face powder without irritating and sensitization.

The Personal Care Products Council (2018) survey of ingredient concentrations used in cosmetics reported that the safety assessments ranged from 110-7% (Palmitoyl tripeptide-1 and Palmitoyl hexapeptide-12) to 0.002% (Palmitoyl hexapeptide-12) [80]. Additionally, the data indicated that these peptides are being used in concentrations between 1 and 30 ppm, and their application at amounts lower than 10 ppm is customary [80]. In fact, a number of commercialized products have been reported to be safe when containing adequate amounts of bioactive peptides. For example, a trade-name formulation containing 100 ppm Palmitoyl tripeptide-12 is non-irritating to the eyes of rabbits [80]. The result of an in-vitro neutral red release assay indicates that a commercialized mixture containing 200 ppm Palmitoyl hexapeptide-12 induces unimportant cytotoxicity [80]. On the other hand, excessive amounts of some peptides have been demonstrated to cause certain side effects. A trade name mixture containing 100 ppm Palmitoyl tripeptide-1 is slightly irritating to the eyes in rabbits. Another trade-name formulation containing 500 ppm Palmitoyl hexapeptide-7 is classified as a mild irritant [80].

The use of Palmitoyl pentapeptide-3 and Palmitoyl hexapeptide-14 for facial powders or spray cosmetics is also regulated at a maximum concentration of 0.06% [4]. Commercialized products named BIOPEPTIDE-CL and NANOFIBERGEL-CS containing 100 ppm Palmitoyl oligopeptide and Palmitoyl dipeptide-18, respectively, were nontoxic in acute oral toxicity studies in rats [4].

As reported by the FDA’s Voluntary Cosmetic Registration Program (VCRP), acetyl hexapeptide-8 is reported to be used in 452 cosmetic products, including 244 leave-on and 30 rinse-off products [81]. Although it is used in concentrations up to 0.005% for eye lotions and face and neck goods, powder products can only contain up to 0.0001% of the total formulations, and a maximum amount of 0.000005% is used for rinse-off (skin cleansing) products [81]. In addition, the report reveals that acetyl hexapeptide-8 can be used as often as several times per day, and that is daily or occasional use may extend over many years without serious side effects.”

Reviewer 2 Report

Beauty peptides have a wide range of biological activities ( anti-aging, antioxidant, anti-inflammatory and antimicrobial activity ) and are ideal candidates for the development of these beauty products. In this review, abundant natural sources ( such as animal, plant and marine sources ) have been identified as the main sources for the extraction of cosmetic peptides through various techniques ( such as enzymatic hydrolysis, ultrafiltration, fermentation and high performance liquid chromatography ). The review of “Insights into Bioactive Peptides in Cosmetics” is very motivating and expressive to readers.

 1.     Peptides composed of short chains of two to fifty amino acids linked by peptide bonds. The contents talking about low allergenicity, allergenicity and discomfort are insufficient.

 2.     In introduction part, the necessity of writing this review is not sufficient.

 3.     Peptides can improve skin health in many ways, including extracellular matrix synthesis, innate immunity, inflammation, and pigmentation, whether specifically targeting common conditions, such as melanin, for example.

 4.     In “table 1”, The paper of “Peptides 2019” was over-cited and the reference format was incorrect.

5.     Other styles errors, like references styles of “Cosmetics 2018”, …

no

Author Response

Response sheet

Manuscript ID: cosmetics-2516626
Type of manuscript: Review
Title: Insights into Bioactive Peptides in Cosmetics
Authors: Le Thi Nhu Ngoc, Ju-Young Moon*, Young-Chul Lee*

Reviewer 2

Beauty peptides have a wide range of biological activities ( anti-aging, antioxidant, anti-inflammatory and antimicrobial activity ) and are ideal candidates for the development of these beauty products. In this review, abundant natural sources ( such as animal, plant and marine sources ) have been identified as the main sources for the extraction of cosmetic peptides through various techniques ( such as enzymatic hydrolysis, ultrafiltration, fermentation and high performance liquid chromatography ). The review of “Insights into Bioactive Peptides in Cosmetics” is very motivating and expressive to readers.

  1. Peptides composed of short chains of two to fifty amino acids linked by peptide bonds. The contents talking about low allergenicity, allergenicity and discomfort are insufficient.

Response: Thank you for your comment. The phrase has been upgraded to describe in more detail the definition of peptides.

“Peptides are short chains of two to fifty amino acids linked together by peptide bonds [1–3]. Amino acids are the building blocks of proteins, and when they are joined in a chain, they form a peptide [1,2]. Peptides are smaller than proteins, typically composed of up to 50 amino acids, although specific thresholds can vary [1–3]. Among the prominent bioactive compounds, peptides have attracted the attention of scientists by their ideal functionalities, especially as regulating/signaling molecules of homeostasis, stress, immunity, defense, growth, and reproduction [1–3], and their own strengths such as high safety, hypo-allergenicity, and their production with cost-effective [1]. Derived from various natural sources such as plants, animals, and microorganisms, these peptides have demonstrated diverse physiological effects, including antioxidant, anti-aging, moisturizing, collagen-stimulating, and wound-healing properties, which have been confirmed through several in-vitro/in-vivo pieces of evidence as well as clinical trial outcomes [1]. For instance, palmitoyl pentapeptide-3 was one of the first synthetic bioactive peptides used to stimulate collagen synthesis for anti-aging and wound healing treatments [4]. The copper Gly-His-Lys (Cu-GHK) was developed and incorporated into cosmetic products to heal skin, promote collagen synthesis, and repair DNA damage [5,6]. Acetyl hexapeptide-3 (Argireline®) is another popular commercialized peptide with potential anti-wrinkle and moisturizing properties [7].”

  1. In introduction part, the necessity of writing this review is not sufficient.

Response: Thank you for your comment. This part has been upgraded to provide more reasonable arguments for the writing this review.

“Despite the growing scientific evidence supporting their potential application in cosmetics, there are only a handful of literature reports describing findings regarding promising applications. For instance, Ferreira et al. (2020) described the utilization of peptides only for anti-aging treatment, not including treatment of all skin conditions [8]. Mazurkiewicz-Pisarek et al. (2023) reported applying antimicrobial peptides only for pharmaceutical, biomedical, and cosmeceutical application [9]. Therefore, documenting the latest research and advancements in a review is necessary to provide comprehensive knowledge that can inspire further innovation in the formulation of peptide-based cosmetic products. In addition, there is a growing consumer demand for safe and effective cosmetic products with natural ingredients or of biological origin. Educating consumers about the potential of bioactive peptides through a well-written paper can help them make informed choices when selecting cosmetic products. It can be seen that understanding the potential of bioactive peptides in cosmetics can have broader implications beyond the cosmetic industry. Therefore, this review aims to explore the vast potential of bioactive peptides for cosmetic applications, shedding light on their intracellular mechanisms of action, their classifications, and their natural origins. It will delve into their multifaceted roles in skin health and beauty, addressing the underlying scientific principles and presenting relevant empirical evidence from both in-vitro and in-vivo studies. Moreover, it will discuss the safety assessment and challenges associated with using bioactive peptides in cosmetic formulations.”

  1. Peptides can improve skin health in many ways, including extracellular matrix synthesis, innate immunity, inflammation, and pigmentation, whether specifically targeting common conditions, such as melanin, for example.s

Response: Thank you for your meaningful comments. This manuscript already described the mechanisms of bioactive peptides in enhancing the skin's appearance. Although peptides can beneficially affect specific common conditions (e.g., melanin, wrinkles, and dehydration) in many ways (e.g., extracellular matrix synthesis, inflammation, and pigmentation), this review is unable to describe the efficacy properties of peptides that correspond to each specific skin condition. Since the skin has undergone so many diseases and disorders, it is hard to present the activity mechanisms of peptides for each skin condition. Therefore, their mechanisms have been described corresponding to each cosmetic property (e.g., antioxidant, anti-inflammation, antimicrobial, and anti-aging activities), as presented in Table 1, to provide concise and complete knowledge of the mechanism of peptides in cosmetic applications.

  1. In “table 1”, The paper of “Peptides 2019” was over-cited and the reference format was incorrect.

Response: Thank you for your comment. The “Peptide 2019” reference has been removed and replaced by other appropriate references so as not to be over-cited. Please see Table 1. Moreover, the reference format has also been corrected according to the journal template.

  1. Other styles errors, like references styles of “Cosmetics 2018”, …

Response: Thank you for your suggestion. All references have been checked and corrected in accordance with the journal template.

Reviewer 3 Report

This paper is interesting and new. Please find my specific comments 

Abstract looks general. Please highlight salient findings.

Keywords: Avoid the words used in title

The introduction section requires more detail to justify the novelty of the work. The  authors should consider providing a more thorough literature review to demonstrate how this review builds upon previous studies in the field. Additionally, the authors should clearly state the research gap that their study aims to fill, and explain why this gap is important to address. By providing a clearer justification for the novelty of the work, the authors will help to engage the reader and increase the impact of their study

The basic information's are explained in detail. It can be shortened.

The recent advances should be highlighted instead of basic details.

The discussion part should be strengthen. Authors have highlighted the results but the discussion part should be improved.

Authors should add a section Methodology to highlight the Keywords used, search engine used, number of articles collected, criteria used to shortlist the articles, etc.,

I recommend the author to include to meta-analysis. It increase the scientific merit of this paper .

Update all the old references

Please carefully revise the manuscript for the appropriate use of definite and indefinite articles (the, and a/an).

Author Response

Response sheet

Manuscript ID: cosmetics-2516626
Type of manuscript: Review
Title: Insights into Bioactive Peptides in Cosmetics
Authors: Le Thi Nhu Ngoc, Ju-Young Moon*, Young-Chul Lee*

Reviewer 3

This paper is interesting and new. Please find my specific comments 

  1. Abstract looks general. Please highlight salient findings.

Response: Thank you for your suggestions. The abstract has been re-phrased to highlight the main content of the manuscript.

“Bioactive peptides have gained significant attention in the cosmetic industry due to their potential for enhancing skin health and beauty. These small protein fragments exhibit various biological activities, such as antioxidant, anti-aging, anti-inflammatory, and anti-microbial activities, making them ideal ingredients for cosmetic formulations. This review provides insight into applying of bioactive peptides in cosmetics and their mechanisms of action (e.g., downregulating pro-inflammatory cytokines, radical scavenging, inhibiting collagen, tyrosinase, and elastase synthesis). Corresponding to the mechanism, bioactive peptides are classified into four categories: signal, carrier, neurotransmitter-inhibitory, and enzyme-inhibitory peptides. The abundant natural origins (e.g., animals, plants, and marine sources) have been identified as primary sources for extractions of cosmetic peptides through various techniques (e.g., enzymatic hydrolysis, ultrafiltration, fermentation, and high-performance liquid chromatography). Furthermore, the safety and regulatory aspects of using peptides in cosmetics are examined, including potential allergic reactions and regulatory guidelines. Finally, the challenges of peptides in cosmetics are discussed, emphasizing the need for further research to fully harness their potential in enhancing skin health. Overall, this review provides a comprehensive understanding of the application of peptides in cosmetics, shedding light on their transformative role in developing innovative and effective skincare products.”

  1. Keywords: Avoid the words used in title

Response: Thank you for your suggestion. The upgraded are as follows: “Keywords: Classification of peptides; mechanisms of action; natural sources; safety assessment”.

  1. The introduction section requires more detail to justify the novelty of the work. The authors should consider providing a more thorough literature review to demonstrate how this review builds upon previous studies in the field. Additionally, the authors should clearly state the research gap that their study aims to fill, and explain why this gap is important to address. By providing a clearer justification for the novelty of the work, the authors will help to engage the reader and increase the impact of their study.

Response: Thank you for your comments. The introduction has been carefully upgraded to increase the impact of our study.

“There is no denying that cosmetics have become an essential part of our daily routine, especially as women, and the market of cosmetics, especially natural ingredient-based products, is more and more growing with great demand for improving the appearance of consumers. Particularly, the development of novelty cosmetic formulations based on bioactive compounds (e.g., antioxidants, proteins, peptides, and growth factors) with therapeutic and protective functions has quickly expanded that can bring out outstanding effects on human skin such as skin whitening, skin moisturizing, and skin rejuvenation [1].

Peptides are short chains of two to fifty amino acids linked together by peptide bonds [1–3]. Amino acids are the building blocks of proteins, and when they are joined in a chain, they form a peptide [1,2]. Peptides are smaller than proteins, typically composed of up to 50 amino acids, although specific thresholds can vary [1–3]. Among the prominent bioactive compounds, peptides have attracted the attention of scientists by their ideal functionalities, especially as regulating/signaling molecules of homeostasis, stress, immunity, defense, growth, and reproduction [1–3], and their own strengths such as high safety, hypo-allergenicity, and their production with cost-effective [1]. Derived from various natural sources such as plants, animals, and microorganisms, these peptides have demonstrated diverse physiological effects, including antioxidant, anti-aging, moisturizing, collagen-stimulating, and wound-healing properties, which have been confirmed through several in-vitro/in-vivo pieces of evidence as well as clinical trial outcomes [1]. For instance, palmitoyl pentapeptide-3 was one of the first synthetic bioactive peptides used to stimulate collagen synthesis for anti-aging and wound healing treatments [4]. The copper Gly-His-Lys (Cu-GHK) was developed and incorporated into cosmetic products to heal skin, promote collagen synthesis, and repair DNA damage [5,6]. Acetyl hexapeptide-3 (Argireline®) is another popular commercialized peptide with potential anti-wrinkle and moisturizing properties [7].

Despite the growing scientific evidence supporting their potential application in cosmetics, there are only a handful of literature reports describing findings regarding promising applications. For instance, Ferreira et al. (2020) described the utilization of peptides only for anti-aging treatment, not including treatment of all skin conditions [8]. Mazurkiewicz-Pisarek et al. (2023) reported applying antimicrobial peptides only for pharmaceutical, biomedical, and cosmeceutical application [9]. Therefore, documenting the latest research and advancements in a review is necessary to provide comprehensive knowledge that can inspire further innovation in the formulation of peptide-based cosmetic products. In addition, there is a growing consumer demand for safe and effective cosmetic products with natural ingredients or of biological origin. Educating consumers about the potential of bioactive peptides through a well-written paper can help them make informed choices when selecting cosmetic products. It can be seen that understanding the potential of bioactive peptides in cosmetics can have broader implications beyond the cosmetic industry. Therefore, this review aims to explore the vast potential of bioactive peptides for cosmetic applications, shedding light on their intracellular mechanisms of action, their classifications, and their natural origins. It will delve into their multifaceted roles in skin health and beauty, addressing the underlying scientific principles and presenting relevant empirical evidence from both in-vitro and in-vivo studies. Moreover, it will discuss the safety assessment and challenges associated with using bioactive peptides in cosmetic formulations.”

  1. The basic information's are explained in detail. It can be shortened. The recent advances should be highlighted instead of basic details.

Response: Thank you for your suggestions. Recent advances and the basic information on the use of bioactive peptides for cosmetic applications are highlighted. This review focuses on the basic knowledge relevant to bioactive peptides, including mechanism, classification, and natural origins. Recent advances in research are also listed in Table 2, showing recent discoveries of bioactive peptides derived from natural sources. We also plan to design a separate paper to explore the cosmetic formulation containing bioactive peptides and develop the delivery system to improve the skin permeability of bioactive peptides. Clinical trials of cosmetic-based peptides are also described in the next paper. Therefore, the advancement of using bioactive peptides is expected to be highlighted in another paper.

  1. The discussion part should be strengthen. Authors have highlighted the results but the discussion part should be improved.

Response: Thank you for your meaningful comment. This is a review paper so it is unnecessary to insert the discussion section. Instead, the conclusion has been re-phrased to highlight the important content presented in this review.

“In conclusion, exploring bioactive peptides for cosmetic applications has opened up new possibilities in skincare and beauty. This review has provided valuable insights into the potential of peptides as active ingredients in cosmetic formulations. Bioactive peptides extracted from various sources (plants, animals, marine sources, and edible insects) exhibit single or multifunctional cosmetic properties, including anti-aging, antioxidant, anti-inflammatory, and antimicrobial activities. Several in-vitro/in-vivo studies and clinical trials demonstrate that bioactive peptides are effective in enhancing skin appearance in both topical applications and oral administration, such as improving skin whitening and moisturizing and reducing skin aging based on various intracellular mechanisms. Safety and regulatory aspects have also been discussed, acknowledging the importance of evaluating potential allergic reactions and complying with regulatory guidelines. Consumer safety is paramount, and thorough testing and assessment must be conducted to ensure the suitability and tolerability of bioactive peptides in cosmetics. In addition, further studies should be conducted to unlock the full potential of bioactive peptides in cosmetics. This includes investigating novel sources of peptides, optimizing extraction methods, and exploring innovative delivery systems to enhance their penetration and activity within the skin. In conclusion, by harnessing their uniqueness to develop innovative and effective skincare products that cater to the diverse needs of consumers.”

  1. Authors should add a section Methodology to highlight the Keywords used, search engine used, number of articles collected, criteria used to shortlist the articles, etc., I recommend the author to include to meta-analysis. It increase the scientific merit of this paper .

Response: Thank you for your meaningful comment. It needs a more specific title, such as on the application of specific peptides for improving specific skin conditions, to perform a meta-analysis study. According to the specific title, we can identify the main objectives and then conduct a literature search to summarize appropriate data for that topic. Therefore, it may be impossible to designa systematic review and meta-analysis paper on this topic. Following your suggestion, a meta-analysis will be designed in the future with a specific topic.

  1. Update all the old references

Response: Thank you for your suggestion. The old references have been replaced by the latest references. Please check the reference section.

Reviewer 4 Report

Thank you for an interesting review of bioactive peptides in cosmetics. This is a detailed review of various aspects of bioactive peptides, including origin, intracellular mechanism, and application. Some minor suggestions: 

  • I like the order of the various areas as they appear in the abstract. I wonder if the authors would consider following the same order in the main body of the review in order to be consistent. Presently the intracellular mechanism section appears before the natural origin section. Or alternatively, change the order in the abstract.

  • Divide section 3 into subheadings similar to Table 2 (which is how it is written).

  • Edible insects do not feature in Table 2

  • Remove the reference to “overview of the safety assessment” from the abstract, as it does not feature in the review 

Specific comments and questions:

“A pseudo-randomized clinical study of 10 Caucasian women confirmed the superiority of soybean peptide (2%) emulsion in terms of collagen and stimulation of glycosaminoglycan contents [33]. “

  • What is the effect on collagen?

“The first carrier peptide was engineered to provide essential wound healing cofactors to the active site efficiently and physiologically for pharmaceutical applications.”

  • Was this engineered for a specific enzyme? Which active site target does this sentence refer to?

“Enzyme-inhibitory peptides can directly or indirectly inhibit enzymes that can reduce the breakdown of collagen and other proteins by interfering with that process”

  • It seems that if these peptides interfere with enzymes that reduce the breakdown of collagen, this is not a beneficial property as you would want to reduce the breakdown of collagen. Please clarify this sentence. 

“In which, common antimicrobial peptides comprise thionins, defensins, 2S albumin-loke proteins, cyclotides, and lipid transfer proteins, of which thionins are the first identified to play a significant role in protecting plants against invading bacteria [38]. 

  • Delete “In which”

Author Response

Response sheet

Manuscript ID: cosmetics-2516626
Type of manuscript: Review
Title: Insights into Bioactive Peptides in Cosmetics
Authors: Le Thi Nhu Ngoc, Ju-Young Moon*, Young-Chul Lee*

Reviewer 4

Thank you for an interesting review of bioactive peptides in cosmetics. This is a detailed review of various aspects of bioactive peptides, including origin, intracellular mechanism, and application. Some minor suggestions: 

  1. I like the order of the various areas as they appear in the abstract. I wonder if the authors would consider following the same order in the main body of the review in order to be consistent. Presently the intracellular mechanism section appears before the natural origin section. Or alternatively, change the order in the abstract.

Response: Thank you for your suggestion. The order of the whole abstract has been rearranged, and the structure of the abstract has also been re-ordered in accordance with the structure of the body.

  1. Divide section 3 into subheadings similar to Table 2 (which is how it is written).

Response: Thank you for your suggestion. Section 3 has been divided into 3 subheadings according to Table 2. The subheading includes “3.1. Plant sources”; “3.2. Animal sources”; “3.3. Marine sources”; “3.4. Edible insect sources”.

  1. Edible insects do not feature in Table 2

Response: Thank you for your comment. The origin of edible insect peptides has been inserted in Table 2.

  1. Remove the reference to “overview of the safety assessment” from the abstract, as it does not feature in the review 

Response: Thank you for your comment. The safety assessment content has been inserted in this upgraded manuscript. Therefore, the phrase “safety assessment” is still presented in the abstract.

  1. Specific comments and questions:

“A pseudo-randomized clinical study of 10 Caucasian women confirmed the superiority of soybean peptide (2%) emulsion in terms of collagen and stimulation of glycosaminoglycan contents [33].What is the effect on collagen?

Response: Thank you for your comment. The phrase has been corrected to be more clarify.

“A pseudo-randomized clinical study of 10 Caucasian women confirmed the superiority of soybean peptide (2%) emulsion in increasing the amount of extracellular matrix components such as collagen and glycosaminoglycan contents”.

  1. “The first carrier peptide was engineered to provide essential wound healing cofactors to the active site efficiently and physiologically for pharmaceutical applications.” Was this engineered for a specific enzyme? Which active site target does this sentence refer to?

Response: Thank you for your comment. The first commercialized carrier peptide was designed to deliver copper, a trace element necessary for wound healing, into the wounded tissue. Therefore, the target active site in this sentence is for promoting regular collagen, elastin, glycosaminoglycan, and proteoglycan synthesis. According to your comment, the phrase has been upgraded to improve clarity.

“Carrier peptides have been designed to deliver essential wound healing cofactors for enzymatic processing and wound repair (Table A1) [10,15]. The first commercialized carrier peptide was designed to deliver copper, a trace element necessary for wound healing, into the wounded tissue.”

  1. “Enzyme-inhibitory peptides can directly or indirectly inhibit enzymes that can reduce the breakdown of collagen and other proteins by interfering with that process”. It seems that if these peptides interfere with enzymes that reduce the breakdown of collagen, this is not a beneficial property as you would want to reduce the breakdown of collagen. Please clarify this sentence. 

Response: Thank you for your comment. Enzyme-inhibitory peptides are used to inhibit the activity of enzymes that break down collagen and other proteins. Therefore, the sentence has been corrected to be more clarify.

“Enzyme-inhibitory peptides can directly or indirectly inhibit enzymes that break down of collagen and other proteins by interfering with that process.”

  1. “In which, common antimicrobial peptides comprise thionins, defensins, 2S albumin-loke proteins, cyclotides, and lipid transfer proteins, of which thionins are the first identified to play a significant role in protecting plants against invading bacteria [38]. Delete “In which”

Response: Thank you for your suggestion. The term “in which” has been removed from the phrase.

Round 2

Reviewer 2 Report

No comments and suggestions.

Reviewer 3 Report

No more comments. Authors revised as per my suggestions. Accepted.